# Spasticity Management after Spinal Cord Injury: The Here and Now

**DOI:** 10.3390/jpm12050808

**Published:** 2022-05-17

**Authors:** Zackery J. Billington, Austin M. Henke, David R. Gater

**Affiliations:** 1Department of Physical Medicine and Rehabilitation, University of Miami Miller School of Medicine, Miami, FL 33136, USA; zbillington@pennstatehealth.psu.edu (Z.J.B.); austin.henke812@gmail.com (A.M.H.); 2Christine E. Lynn Rehabilitation Center for the Miami Project to Cure Paralysis, Miami, FL 33136, USA; 3The Miami Project to Cure Paralysis, University of Miami Miller School of Medicine, Miami, FL 33136, USA

**Keywords:** spinal cord injury, spasticity, spasms, hyperreflexia, neurolysis

## Abstract

Spasticity is a common comorbidity of spinal cord injury (SCI) that is characterized by velocity dependent tone and spasms manifested by uninhibited reflex activity of muscles below the level of injury. For some, spasticity can be beneficial and facilitate functional standing, transfers, and some activities of daily living. For others, it may be problematic, painful, and interfere with mobility and function. This manuscript will address the anatomy and physiology of neuromuscular reflexes as well as the pathophysiology that occurs after SCI. Spasticity assessment will be discussed in terms of clinical history and findings on physical examinations, including responses to passive and active movement, deep tendon reflexes, and other long tract signs of upper motor neuron injury, as well as gait and function. Management strategies will be discussed including stretch, modalities, pharmacotherapy, neurolysis, and surgical options.

## 1. Introduction

Spasticity has been commonly defined as a velocity dependent increase in muscle tone due to exaggerated stretch reflexes [1]. Some have described this definition as “too limiting,” and an alternate definition has been gaining favor, defining spasticity as “disordered sensorimotor control, resulting from an upper motor neuron lesion, presenting as intermittent or sustained involuntary activation of muscles” [2]. Spasticity is a fairly common, often underdiagnosed, and important symptom of upper motor neuron (UMN) syndrome in persons with spinal cord injury (SCI). In the early 1990s, Maynard et al. reported 67% of individuals with new SCI had some degree of spasticity at the time of discharge to the community, including 89% of those with tetraplegia, 82% of those with T1–T7 paraplegia, 45% of those with T8–T12 paraplegia, and 26% of individuals with L1–L5 paraplegia [3]. While only 26% of those individuals required treatment at the time of discharge, 46% reported problematic spasticity requiring medical management at 1-year follow up. Recent literature demonstrated similar findings, with 65% of persons with traumatic SCI reporting spasticity at the time of discharge, 35% of whom required medical management for problematic spasticity [4]. On follow up, problematic spasticity requiring medical management was noted in 35% of those individuals at 1 year, 41% at 2 years, and 31% at 5 years after their initial SCI. Skold et al. performed a survey to help further determine the prevalence of spasticity by ASIA grade and neurologic level of injury [5]. Three hundred fifty-four spinal cord injured individuals participated in the survey. The data collected was analyzed based on participants’ grade and general level of injury (cervical, thoracic, and lumbosacral). Survey data showed participant-described “problematic” spasticity was most prominent in cervical neurologic levels of injury, particularly in incomplete ASIA grades (ASIA grades B–D). In comparing problematic spasticity by levels, the data collected showed 79% of cervical injuries, 69% of thoracic injuries and 22% of lumbosacral injuries had problematic spasticity [5]. Skoog et al. performed a review of 161 patients in Sweden with spinal cord injury, and found that 87% of cervical level of injury, 85% of thoracic injury and 57% of lumbar injury developed spasticity [6]. Severe spasticity can reduce quality of life, limit functional abilities or social goals, and cause distress and pain for the individual. Understanding spasticity, its clinical features and its management are important skills for SCI professionals to optimally treat and care for their patients.

Sir Charles Scott Sherrington, a neurophysiologist, pathologist, and 1932 Nobel Prize winner for his work on the function of neurons, posited that the nervous system, composed of the brain, spinal cord, and peripheral nerves, coordinated the movements of the body with the utilization of simple reflex pathways to coordinate, plan, and execute the many bodily functions. Sherrington believed reflexes illustrated the simplest components of the nervous system structure and called the lower motor neurons (LMNs) the “final common pathway for reflex arcs,” as these muscle controllers facilitated output from the central nervous system (CNS). Since Sherrington, we have discovered the multi-layered complexity of the nervous system, and still Sherrington’s words ring true. It is the alpha motor neurons, modulated by many different inputs above, that facilitate messages from the CNS to the periphery. Spasticity is a result of damage to the CNS, so to understand spasticity, one must understand the interactions between the CNS and the peripheral nervous system (PNS) it controls. The current manuscript underscores the physiologic neural mechanisms of spasticity, including the LMN cell body, its axon hillock, the muscle spindle, additional excitatory influences, epidemiology, pathophysiology, and treatment modalities utilized in treating spasticity in persons with SCI. 

## 2. Anatomy and Physiology: The Basics

Physiologically, the Ia afferent neuron from the muscle spindle and the alpha (α) motor neuron is, as Sherrington posited, the “final common pathway” of the reflex arc. The muscle spindle is a complex sensory receptor that runs parallel to and inserts into the end sheaths of extrafusal skeletal muscle fibers. It is extremely sensitive to changes in length over time. The spindle is composed of a central nuclear bag and more distal nuclear chain receptors that activate group Ia and smaller group II afferent neurons respectively, in response to changes in muscle length (Figure 1) [7]. Under conditions of stretch (such as a reflex hammer elicits at the patellar tendon), these afferent neurons synapse in the spinal cord with α motor neurons that activate the extrafusal skeletal muscle fibers of the agonist muscle. Commensurate coactivation of γ motor neurons to intrafusal muscle fibers at each end of the spindle maintains the sensitivity of the muscle spindle as the extrafusal fibers contract and the muscle shortens [8].

While the alpha motor neuron cell body (soma) lies in the anterior horn of the spinal cord, its axon hillock functions as the origin of the generated α motor neuron action potential. The axon hillock is the site where all mono- and polysynaptic inputs are received and summated from the surrounding dendritic connections and is the transition point where voltage-gated ion channels accumulate positive and negative charges. The resting membrane potential at the axon hillock is −70 mV. Excitatory post-synaptic potentials (EPSPs) respond to temporal and spatial activation by excitatory amino acids such as glutamate, aspartate, and N-methyl aspartate (NMDA) that cause an influx of sodium (Na^+^) into the axon hillock. Inhibitory post-synaptic potentials (IPSPs) respond to temporal and spatial activation by the inhibitory amino acids, gamma aminobutyric acid (GABA) and glycine, causing an influx of chloride ions (CL^−^) into the axon hillock, with an outflow of potassium (K^+^). The summation of these EPSPs and IPSPs creates a change in the membrane potential, and if the threshold for an action potential is reached (−55 mV), an “all-or-none” monosynaptic action potential will be generated, ultimately leading to the excitation contraction coupling necessary for agonist muscle contraction. Of note, Ia reciprocal inhibition through disynaptic inhibitory pathways suppress antagonistic motor units, while Renshaw Cell interneurons through polysynaptic inhibitory pathways may also suppress antagonist motor units (recurrent inhibition).

Interneurons within the spinal cord are neural components that allow supraspinal modulation of reflexes to derive purposeful movement and smooth, coordinated flow of movement. Interneurons constitute up to 90% of the neuronal network within the spinal cord, outnumbering the corresponding motoneurons they modulate by up to 30 times [7]. The spinal cord has many connections interfacing with its substance from subcortical and cortical origins that help to facilitate and modulate the muscle stretch reflex sensitivity, facilitation, and response to extrinsic stimuli. Figure 2 illustrates the many different descending cortical and subcortical inputs onto the spinal cord, the summation of which modulates the response downstream at the muscular level. These connections can be grossly divided into excitatory and inhibitory, and the summation of the potentials from each pathway generates an effect on the spinal cord and muscular reflexes [7]. Five descending tracts in particular modulate reflexive activity via intraspinal interneurons, including the corticospinal, reticulospinal, vestibulospinal, rubrospinal, and tectospinal tracts, with the corticospinal, corticoreticular, and dorsal reticulospinal tracts providing inhibitory influences, while vestibulospinal and medial (ventral) reticulospinal tracts provide excitatory influences on spinal reflexes [7].

## 3. Pathophysiology

Spasticity is a symptom of the over-arching UMN syndrome, and in its simplest form is a loss of supraspinal modulation of muscle stretch reflexes. This differs in comparison to muscular spasms, which are uncontrolled, periodic, non-sustained involuntary muscular contractions, and not part of the UMN syndrome. As discussed above, corticospinal, corticoreticular, and dorsal reticulospinal tracts provide supraspinal inhibitory influences on spinal reflexes in the intact CNS, but once SCI occurs these inhibitory influences are lost to the extent that these tracts are damaged. Recent findings have suggested that a link exists between corticospinal connectivity and spasticity, such that reduced corticospinal activity is associated with higher levels of spasticity [9]. Further, imbalanced corticospinal and reticulospinal influences appear to contribute to spasticity in those with incomplete SCI [10]. Those without significant spasticity show no evidence of motor evoked potentials (by transcranial magnetic stimulation of the motor cortex) and large lateral spinal atrophy, while those with greater spasticity appear to have greater corticospinal responses and less atrophy [11].

Additionally, intrinsic changes occur in the spinal cord after injury, during the “recovery process,” including neuroplasticity and axonal sprouting, as initially posited by McGouch et al. in 1958 [12]. After injury to the spinal cord tissue, the injured neuronal axons begin to die and degenerate, leaving intact axons an opportunity to send sprouting branches out to create new synaptic junctions. An increase in afferent synapses likely causes increased synapses onto interneurons, exerting more of an effect with each potential through that spinal pathway. One consequence of disordered axonal sprouting can be increased synaptic input onto interneurons that are part of a persistent inward current pathway (PIC) [13]. PICs are pathways that continually reach the threshold for action potential propagation despite continued, prolonged depolarization (i.e., no refractory period). PICs are modulated by cortical centers via monoaminergic input onto the pathways to inhibit them. With a spinal cord injury, cortical modulation is lost, and PICs can contribute to developing spasticity [13,14].

Unperceived noxious stimuli below the level of injury can also cause increased spasticity in those with SCI by increasing EPSPs at all final common pathways caudal to the SCI, unopposed by supraspinal inhibition. A recent systematic review identified spasticity triggers from multiple sources including bladder, bowel, pregnancy, posture, cold, menses, fatigue, mental stress, skin conditions, bracing, and tight clothing in persons with SCI [15]. Abrupt worsening or new spasticity should prompt a thorough search by the clinical team to determine the source, and removal of the stimuli should be of paramount importance in management. 

## 4. Assessment

Spasticity is not always problematic. Many individuals with spinal cord injury utilize their spasticity to be more efficient in their activities of daily living—such as with transfers, truncal stability for postural control, and maintenance of blood pressure by reduction in venous pooling. However, it can be problematic and affect an individual’s ability to perform basic functional tasks and activities of daily living, e.g., difficulty with skin integrity, hygiene completion, and even toileting can be impacted by spasticity. Pain is often a component of spasticity and can also lead to reduced functionality, independence, and quality of life. 

A detailed clinical history of how spasticity affects an individual’s function is one of the key assessments for determining appropriate treatment. In usual practice, spasticity is only treated when it causes detriment to the patient, such as pain, functional limitations, formation of muscle contractures, or positional limitations. Muscle contractures in SCI are due to a loss of extensibility of the soft tissue structures spanning joints [16]; some portion of this is believed due to unbalanced spasticity across a joint leading to a reduction of muscle sarcomeres in series [17]. One year after SCI, 66% of persons with SCI had developed at least one joint contracture, including 43% shoulder, 33% elbow and forearm, 41% wrist and hand, 32% hip, 11% knee and 40% ankle contractures [18]. When encountered, a detailed history of functional limitations and their progression is essential to determine treatment intervention.

The quality of the spasticity, its severity, distribution, and any alleviating or aggravating factors are important considerations when devising a treatment plan and may guide the clinician to different modalities depending on patients’ explanations of their spasticity. As above, helping a patient determine potential triggers of their spasticity can assist them to notice more subtle changes which could reflect underlying pathological issues that may need treatment. Identifying potential triggers can help prevent co-morbid illnesses or complications from further reducing quality of life and function of an individual with SCI [19,20]. 

Several clinical assessment tools for spasticity have been developed over the years, including the Modified Ashworth Scale (MAS) [21]; the Modified Tardieu Scale (MTS) [22]; the Penn Spasm Frequency Scale (PSFS) [23]; and the Spinal Cord Assessment Tool for Spastic Reflexes (SCATS) [24]. Table 1 highlights the MAS, MTS and PSFS spasticity scale grading systems. The MAS is a six-point ordinal scale for grading resistance encountered during passive movement stretching at increasing velocity [21]. The MTS has recently been validated in SCI and is felt to be superior to the MAS, as it can distinguish between neural and peripheral contributions to spasticity [22]. The PSFS is a patient self-reporting tool which can provide information on the severity and frequency of spasticity [23]. Finally, the SCATS is preferred by many SCI clinicians [25], and includes ordinal scales for plantar flexor *clonus*, *flexor spasms* and *extensor spasms*. Plantar flexor *clonus* is determined as 0, if no reaction; 1, if mild clonus is present for <3 s; 2, if moderate clonus persists for 3–10 s; and 3 if severe clonus lasting > 10 s is present. Hip, knee, and ankle *flexor spasms* or great toe extension in response to a 1 s pinprick stimulus at the medial arch is scored as 0, if no response; 1, if mild response with < 10° hip, knee and ankle flexion, or great toe extension; 2, if moderate response with 10–30° hip, knee and ankle flexion, or great toe extension; and 3, if severe response occurs with > 30° hip, knee and ankle flexion, or great toe extension. With the contralateral leg extended, *extensor spasms* are measured as 0, if no reaction; 1, if mild patellar clonus is present < 3 s; 2, if moderate patellar clonus persists 3–10 s; and 3, if severe patellar clonus lasting > 10 s is present when hip and knee were simultaneously extended from ~ 100° flexed positions while cupping the heel.

In addition to the tools listed above, the clinical exam should include testing muscle stretch reflexes, commonly referred to as deep tendon reflexes (DTRs) that can help elucidate the severity of hyperreflexia, scoring them with 0 is no response, 1+ is hyporeflexia, 2+ is normal, 3+ is brisk, and 4+ is a reflex that elicits clonus. The exam should include biceps (C5), extensor carpi radialis (C6), triceps (C7), patellar tendon (L2, L3, and L4), and Achilles tendon (S1) reflexes. Other clinical examination maneuvers to uncover signs of an upper motor neuron injury such as the Hoffman reflex, and an upgoing toe in response to a Babinski maneuver and/or crossed hip adductor reflex can confirm an upper motor neuron injury. Functional evaluations such as gait analysis and an active range of motion/strength assessment can assist to quantify the severity and functional impact of spasticity on the patient.

## 5. Management/Treatment

Management of spasticity is no longer performed on a graded or tiered approach, but all modalities and treatment options can be performed in a sequential or synergistic model of treatment to best manage the patient. The distribution of the spasticity can guide what treatment options may be best suited for each patient. In individuals with considerable diffuse spasticity, systemic medical treatments may be of most benefit, whereas focal isolated spasticity may best be treated with local interventions. Presented here is a summary of the treatment options starting from conservative and progressing to more invasive/surgical treatment options. 

### 5.1. Conservative Management

Often in spinal cord injured individuals, spasticity is a manifestation of a noxious stimulus such as UTI, severe constipation, or skin injury/osseous injury. As such, the first management option and often effective treatment of spasticity is removing any noxious stimuli or identifying potential noxious stimuli, which can assist in reducing spasticity. It is recognized that for individuals with SCI at or above T6, such stimuli can also provoke autonomic dysreflexia, a hypertensive crisis that warrants immediate medical attention. This entity is more fully discussed in another manuscript in this special issue. (REF ANS Manuscript)

Initial management of emerging spasticity often begins with range of motion, and massage and stretching techniques utilizing the length/tension relationship of muscle fibers, optimizing the length of overlap of actin and myosin to maintain proper joint position, and reducing unbalanced forces across a joint. Proprioceptive neuromuscular facilitation utilizes agonist/antagonist muscle spindle and golgi tendon organ physiology to provide reciprocal inhibition of muscles across joints to optimize stretch [26]. Bracing, taping, and casting also utilize proper joint position and consistent, constant muscular stretch in order to maintain or optimize proper muscle fiber length, joint position, or functional force generation, and may be used in conjunction with the focal therapies discussed below.

### 5.2. Therapeutic Modalities

There are several temperature-based modalities utilized to attempt to reduce spasticity in spinal cord injury. Cryotherapy, the utilization of cold temperature modalities, reduces spasticity by reducing muscle spindle afferent sensitivity to muscle contraction and stretch; however, the efficacy is short lived and transient in nature. Thermotherapy works to reduce spasticity in a transient manner by causing relaxation or increased elasticity of the extracellular matrix and may also reduce afferent spindle sensitivity. Neither of the temperature-based modalities have significant evidence-based research demonstrating long-term benefit in persons with SCI. Transcutaneous electrical neuromuscular stimulation (TENS) may be useful to reduce painful stimuli that provoke spasticity below the level of injury, and one study demonstrated similar efficacy as oral baclofen in reducing the Ashworth score, PSFS and ankle clonus [27]. Transcutaneous spinal cord epidural stimulation has been trialed in several studies, but with mixed results [28]. Functional electrical stimulation leg cycle ergometry has also been deemed efficacious in reducing spasticity in persons with various levels of SCI, as reported in a recent systematic review [29]. These biomechanical modalities in combination with physical therapy can assist in modulating spasticity and have some efficacy in spasticity treatment. However, for many cases, therapy and biomechanical modalities need additional pharmacologic interventions to adequately treat spasticity in persons with SCI [19,20,30]. 

### 5.3. Oral Pharmacological Management

Most pharmacologic spasticity treatment options work in the central nervous system on interneuronal or motoneuronal spinal pathways. The most common side effects of the anti-spasmodics are sedating qualities with their action on the central nervous system. These medications are usually very good options in cases of generalized spasticity, as they are systemically active, but they may be perceived as causing muscle weakness [19].

#### 5.3.1. Gaba Agonists

Pharmacotherapy takes advantage of what has been learned about the activation of nerve potentials at the axon hillock. GABA agonists increase CL- inflow and K+ outflow to hyperpolarize the hillock, increasing IPSPs and reducing hyperreflexia. The two main GABA agonists utilized for spasticity are baclofen and diazepam. Baclofen is a GABA_B_ receptor agonist and is probably the most commonly prescribed antispasmodic in all patient populations dealing with spasticity, especially those with SCI [19,31]. Baclofen binds to the GABA_B_ pre- and post-synaptically, causing inhibition of both monosynaptic and polysynaptic reflexes. Presynaptic binding also causes a reduction in the influx of calcium into the presynaptic terminal, additionally reducing alpha motor neuron activity. Baclofen crosses the blood brain barrier and is metabolized by the liver and renally excreted; those with concomitant kidney disease should be renally dosed or tapered off the medication. Side effects of baclofen include sedation, drowsiness, ataxia, and respiratory and cardiovascular depression. Overdose symptoms include weakness, areflexia, hypotonia, respiratory depression, seizures, and death. Of note, abrupt withdrawal of baclofen may cause rebound spasticity, motor hyperactivity, headache, insomnia, hallucinations, seizures, and fever [19,31,32]. Diazepam, a GABA_A_ receptor agonist, also acts pre- and post-synaptically, inhibiting both monosynaptic and polysynaptic reflexes, and like baclofen, increases IPSPs to reduce spasticity. Diazepam side effects include sedation, drowsiness, attention impairment, and memory impairment, which may limit its use, but may make it an excellent option for problematic nocturnal spasticity. There is always a risk for respiratory depression, drug tolerance and dependency with increased doses [19,32].

Of note, gabapentin is sometimes used to manage spasticity, but its mechanism of action is likely to be due to its analgesic effects on neuropathic pain and subsequent reduction of EPSPs at the axon hillock. Although the medication is structurally similar to GABA, it does not activate GABA receptors [33].

#### 5.3.2. Alpha Agonists

Alpha-2 (α_2_) adrenergic agonists such as clonidine and tizanidine work pre-synaptically to reduce the release of glutamate, aspartate, and NMDA from spinal interneurons and their excitatory influences on the α motor neuron cell body, i.e., decreasing EPSPs at the axon hillock [19,32]. Clonidine is also a potent antihypertensive agent, and its side effects of hypotension, bradycardia, sedation, and dizziness limit its usefulness in persons with SCI unless they have concomitant hypertension. There is also a risk of withdrawal symptoms if the medication is abruptly stopped. Tizanidine, also an α2 adrenergic agonist, has similar effects on spasticity as clonidine, but with significantly fewer side effects of hypotension, bradycardia, and sedation due to its relatively limited effects on the cardiovascular system [19,32]. Liver function tests (LFTs) should be performed prior to initiation and 4–6 weeks afterward as there is a risk for hepatotoxicity [32]. Of note, it can be effective when used in combination with baclofen, as its mechanism of action is different. 

#### 5.3.3. Calcium Antagonists

Dantrolene works peripherally on the muscle fibers, binding to the ryanodine receptors and reducing the calcium release from the sarcoplasmic reticulum, uncoupling the excitation–contraction mechanism [19,32]. While effective on central systemic spasticity, it also weakens muscle that is under voluntary control and therefore may not be the best choice for managing spasticity from SCI. There is a subset of individuals that have hepatotoxicity, and mild liver enzyme elevations are common, so LFTs should be checked prior to initiation of the medication and 4–6 weeks later [32]. 

## 6. Intrathecal Baclofen

In addition to oral baclofen therapy, intrathecal administration of baclofen (ITB) is considered in individuals with spasticity that may have been inadequately controlled by oral medications alone, or in patients in whom oral administration was effective but side effects limited increases in dosage. ITB is administered continuously via an internal pump, reservoir, and catheter system that is programmed to deliver baclofen into the thecal space, essentially bathing the spinal cord with medication. By delivering concentrated medication directly to the spinal cord, the provider is able to drastically reduce the dosage (ITB dosed in micrograms), thereby limiting side effects [34,35,36,37]. The system can be assessed intermittently to optimize dosing administration, and refills can be extended to several months, depending on the concentration of baclofen placed in the reservoir. Minor surgery is required to place the pump, and although infrequent, mechanical failures can happen, leading to overdose or withdrawal symptoms [34,35,36,37]. An essential element of managing ITB pumps is troubleshooting either loss of ITB efficacy or ITB overdose, both of which can appear as life-threatening circumstances [36].

## 7. Neurolysis

### 7.1. Botulinim Toxin Injection

Chemodenervation with botulinum toxin (BoNT) injections offers more localized, muscle-specific anti-spasmodic treatment. In persons with SCI who have isolated spasticity or are on systemic treatment but still have some degree of focal difficulty, BoNT provides an adjunctive treatment option. BoNT, derived from the *Clostridium Botulinum* bacterium, exerts its influence on the pre-synaptic motor nerve terminal by blocking or reducing the amount of acetylcholine release into the neuromuscular junction. The toxin blocks the synaptosomal-associated receptor (SNARE) complex binding to the axon terminal wall and preventing exocytosis of the acetylcholine vesicle [38,39]. There are many different serotypes (A-G) of toxin that work on different components of the SNARE complex; the three most commonly used are onabotulinum toxin A (Botox^®^), abobotulinum toxin A (Dysport^®^), and incobotulinum toxin A (Xeomin^®^). The toxin is reconstituted in saline and injected intramuscularly into the spastic muscles, sometimes with the aid of a neurostimulator. Patients should be educated on the potential spread of the medication and adverse side effects such as allergy to the toxin, injection site reactions, and if injecting near the neck, potential for spread to unintended muscles, creating transient breathing or swallowing difficulties [38,39]. 

### 7.2. Phenol Neurolysis

Another useful tool is chemical neurolysis of specific motor nerves with phenol or concentrated ethyl alcohol, using imaging guidance with ultrasound or motor point blocks with electromyography (EMG) or neurostimulation guidance. These concentrated caustic agents cause focal demyelination and destruction of the nerve and surrounding tissue, causing reduction in tone by breaking the reflex arc. Of note, administration should only be provided to pure motor nerve branches, e.g., the musculocutaneous nerve to the biceps or the obturator nerve to the hip adductors, as it can cause significant dysesthesias in mixed or sensory nerves; motor point blocks may be as effective and less invasive [19]. Phenol and ethyl alcohol administered in the nerve trunk cause a transient effect, similar to lidocaine administration, and are utilized mostly in pre-operative diagnostics to assess for therapeutic effect after surgery. Chemical neurolytics are not utilized as often in spinal cord injury due to the variable duration of effect, lasting from three to eight months [13,19]. Due to its caustic and denaturing mechanism of action, side effects from administration can range from injection site reactions, allergy to medication, dysesthesias if a mixed nerve is targeted, permanent palsy of the nerve with repeated injections, potential damage to vascular, skin and subcutaneous tissues, and phlebitis; if injected into circulation, systemic side effects as severe as cardiovascular collapse can occur [13,19]. 

## 8. Non-Pharmacological Neuromodulation

Transcranial magnetic stimulation (TMS) is a non-pharmacologic neuromodulation treatment being studied for spasticity. Electro-magnetic fields generated by the treatment cause depolarization of a selected pool of neurons. Depending on frequency, the amplitude and pattern of TMS can assist in spasticity management. Benito et al. found that TMS performed on incomplete spinal cord injury led to a 41% reduction in MAS scores [40]. Extra-corporeal shock wave therapy (ESWT) is a modality utilizing sonic pulses at various frequencies and pressures to induce rheological changes in the muscle tissue itself, thereby reducing tone and treating spasticity. It is also believed that ESWT induces biological responses, such as increased angiogenesis factors and nitric oxide. Some studies have shown a reduction in MAS and improvement in joint range of motion [41,42,43].

## 9. Surgical Considerations

In severe, refractory cases, surgical options exist to reduce spasticity by attempting to break the reflex feedback loop. Selective neurotomy, i.e., transecting a specific nerve to decrease spastic muscles, may be necessary and advisable in some circumstances. For example, obturator neurotomy may be used to reduce adductor tone that otherwise prevents perineal hygiene [19]. Dorsal rhizotomy, i.e., radiofrequency ablation of dorsal rootlets, reduces spasticity by reducing afferent input onto the spinal cord, and thus breaking the reflex cycle [13,19]. Rhizotomy causes some anesthesia at the dermatomal level supplied by the rootlets. Myelotomy, surgical incision (usually midline) of the spinal cord, is primarily utilized for pain control; however, in reducing sensory input, it can help to reduce spasticity as well in a manner similar to rhizotomy in breaking the reflex arc [13]. Cordotomy works in a similar fashion, however, has a different entry point. All of these surgical procedures work to reduce spasticity by affecting the reflex arc. They are permanent procedures, and there may be some associated loss of sensation and, in some cases, function due to these procedures, which cannot be reversed. However, in some refractory cases, surgical consideration may be needed to ultimately meet the patient’s quality of life and functional goals.

## 10. Conclusions

Spasticity is a debilitating and sometimes misunderstood and undiagnosed symptom of an upper motor neuron injury. It can cause functional limitations, pain, social anxiety, and distress to the individual. It is important for the SCI physician to act as an advocate for education and understanding of the condition. Adequate diagnosis, clinical examination, and understanding established treatment options can optimize patient care. With further research and technology focused on spasticity, new treatment options are being discovered to improve quality of life, function, and overall social reintegration for persons with SCI. 

## Figures and Tables

**Figure 1 jpm-12-00808-f001:**
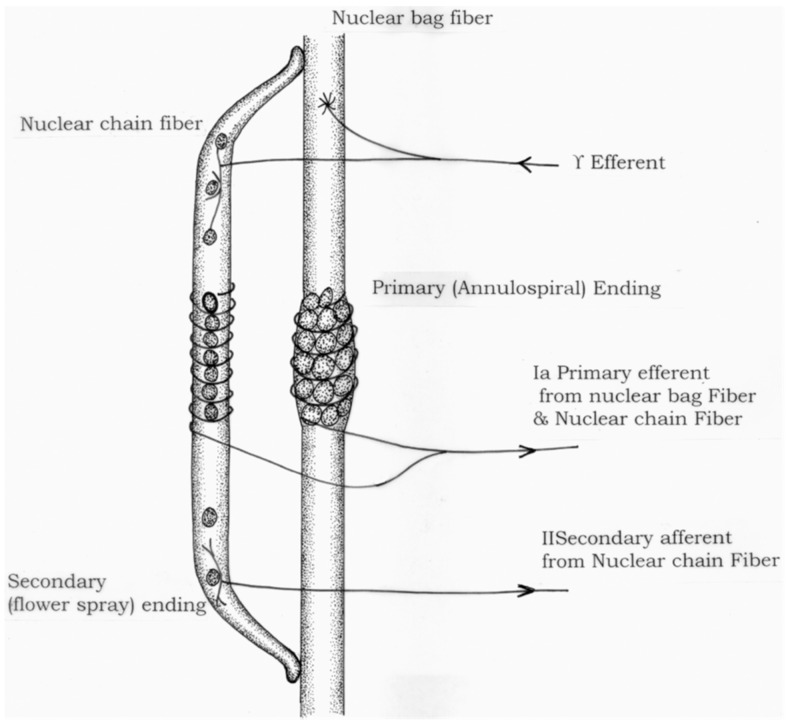
Diagrammatic representation of a muscle spindle. Adapted from Ref. [7].

**Figure 2 jpm-12-00808-f002:**
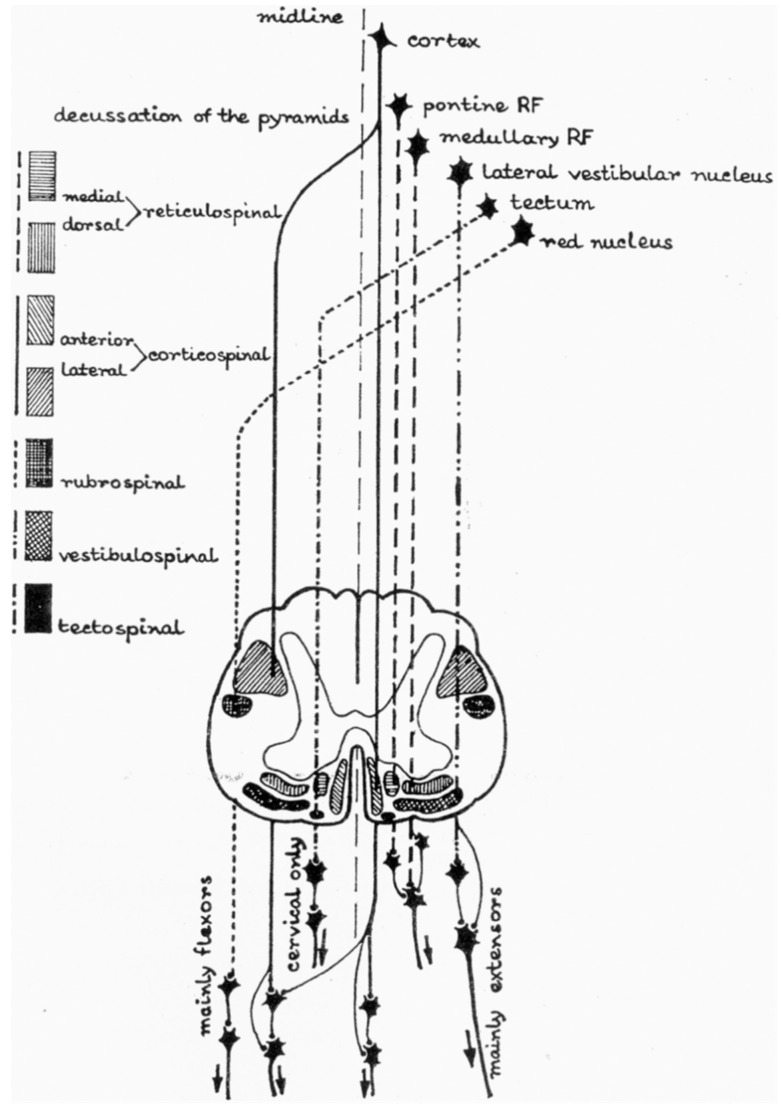
Supraspinal descending pathways in the spinal cord (RF: reticular formation). Adapted from Ref. [7].

**Table 1 jpm-12-00808-t001:** Spasticity Scales.

Grade	Spasticity Scale			Deep Tendon Reflexes	
	Modified Tardieu Scale	Modified Ashworth Scale	Penn Spasm Frequency Scale		
0	No resistance throughout the course of passive movement	No increase in tone	No spontaneous spasms	1+	Hyporeflexic response
1	Slight resistance through the course of passive movement, with no clear catch	Slight increase in tone, manifested by a catch and release or minimal resistance at end range of motion	No spontaneous spasms; spasms with rigorous sensory or motor stimulation	2+	Normal reflex response
2	Clear catch interrupting the passive movement, followed by a release	Marked increase in tone, manifested by a catch at mid-range of motion and resistance throughout remainder of motion	Occasional spontaneous spasms and easy induced spasms	3+	Brisk reflex response—no clonus elicited
3	Fatigable clonus (<10 s) while maintaining pressure	Considerable increase in muscle tone, with passive movement difficult	>1 but <10 spontaneous spams in an hour	4+	Brisk reflex response with associated clonus
4	Infatiguable clonus (>10 s) while maintaining pressure	Affect part rigid and fixed in place	>10 spontaneous spasms in an hour		

## Data Availability

Not Applicable.

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
