# Peer review of "Spasticity Management after Spinal Cord Injury: The Here and Now"

_jpm, 2022, doi:10.3390/jpm12050808_

Round 1

Reviewer 1 Report

Review of jpm 1708378, Spasticity after Spinal cord injury (SCI)

This paper is a comprehensive review at a readily understandable level, ideal for students, trainees and those working in rehabilitation. The review covers the neurological basis, clinical and lifestyle effects, and available treatments for spasticity following spinal cord injury. It is well written and easy to follow.   

While the two illustrations, and many of the concepts described in the text are already described in 2010 in reference 6 (Mukherjee & Chakravarty), in the present paper the authors expand particularly on the mechanism and range of therapeutic options. 

Main points.
1. Lines 37-40  ; Lines 160-163.  The paper quotes figures for percentages of spasticity at different levels, and for percentages with contractures (after 1 yr) at different joints.  The authors need to present data also on the distributional proportion of patients with SCI at each anatomical spinal cord level, and ideally express the quoted percentages against each SCI level.  Only then can one gauge how the quoted percentage data can be applied prospectively to patients with SCI at a particular level.   

  1. Lines 172-201; and 202-205. The authors should consider whether the grading of several of the clinical assessment tools might be better presented in a combined Table of Assessment Tool as columns and Score as rows; filling in the relevant cells for each assessment tool with the clinical property for that score.  

More detailed comments are mostly a preference for some minor alterations to the grammar in occasional sentences.  These are:

  1. Line 50. ….pathologist who won the 1932 …. Suggest delete ‘who’   Alternatively delete the ‘. He’ at the start of the following sentence to make a single sentence, but this would be cumbersomely long.
  2. Line 107-111 Please cite the reference to the ‘5 descending tracts’ even if this is also reference 6.
  3. Lines 226 ‘…stimuli can…’ Should be:  ‘…stimuli, and can…’
  4. Line 242 ‘…sensitivity to spasticity and stretch…’    Do the authors mean ‘…stimulation…’ or ‘…muscle contraction…’ here rather than …’spasticity…’
  5. Line 283 ‘…attention impairment, memory impairment…’ Better adding ‘…and…’   ie. ‘…attention impairment and memory impairment…’
  6. Line 286-7 ‘…likely due to…’ Better as  ‘…likely to be due to…’
  7. Line 288-9 Gabapentin is a correct chemical name, so not a 'misnomer'. It is a derivative of GABA with a pentyl disubstitution at the 3 position; and so it is irrelevant for its name whether it is a GABA agonist or a competitive blocker (either of which could be the case for a chemical derivative of a physiologically active substance). I suggest deleting the suggestion of it being a ‘misnomer’.
  8. Line 308-9 ‘…it also weakens muscle that are……and therefor may not be…’  Do the authors wish ‘muscle’ to be singular or plural.  This would read better as :  ‘…it also weakens muscle that is……and therefore may not be…’
  9. Line 331 ‘Chemodenervation with…… injections offer…’ This needs to be singular:  ‘Chemodenervation with….  injections offers
  10. Line 336 This might be hyphenated: ‘….synaptosomal-associated….’
  11. Line 341 ‘…The toxin in reconstituted.…’   Should be ‘..is..’ rather than ‘..in..’
  12. Line 348 ‘…neurolysis with…’ It may be better to clarify that here as : ‘…neurolysis of specific motor nerves with….’

Reviewer 2 Report

This review is very well written and interesting.

Nevertheless, a few non pharmacological neuromodulation options are not detailed. Indeed these options are rather new but they are very promising and should be presented, as in a recent review (Hodge JO, Brandmeir CL, Brandmeir NJ. Neuromodulation Therapies for Spasticity Control: Now and Beyond. Neurol India 2020;68, Suppl S2:241-8).

In particular, the noninvasive repetitive Magnetic Transcranial Stimulation (Nardone R et al. Restor Neurol Neurosci. 2017;35(3):287-294; Korzhova J et al. J Phys Rehabil Med. 2018 Feb;54(1):75-84 ); extracorpora Shock Wave Therapy (Martinez IM et al.Brain Sci. 2020;11(1):15 ; Khan F et al.Ann Phys Rehabil Med. 2019;62(4):265-273).

At last, the reader should be more aware that SCI-induced spasticity is evolving rapidly, as gross motor issues themselves are starting to be solved (Wagner FB et al. Nature. 2018;563(7729):65-71 ; Hachmann JT et al. J Neurophysiol. 2021;126(6):1843-1859). Otherwise, it is a good review article.
